# DNA Methylation, Mechanisms of *FMR1* Inactivation and Therapeutic Perspectives for Fragile X Syndrome

**DOI:** 10.3390/biom11020296

**Published:** 2021-02-16

**Authors:** Veronica Nobile, Cecilia Pucci, Pietro Chiurazzi, Giovanni Neri, Elisabetta Tabolacci

**Affiliations:** 1Sezione di Medicina Genomica, Dipartimento Scienze della Vita e Sanità Pubblica, Fondazione Policlinico Universitario A. Gemelli IRCCS, Università Cattolica del Sacro Cuore, 00168 Rome, Italy; veronicanobile88@gmail.com (V.N.); c.pucci91@yahoo.it (C.P.); pietro.chiurazzi@unicatt.it (P.C.); giovanni.neri43@gmail.com (G.N.); 2Fondazione Policlinico Universitario A. Gemelli IRCCS, UOC Genetica Medica, 00168 Rome, Italy; 3Greenwood Genetic Center, JC Self Research Institute, Greenwood, SC 29646, USA

**Keywords:** fragile X syndrome, *FMR1* gene, epigenetic modifications, DNA methylation, gene expression signatures

## Abstract

Among the inherited causes of intellectual disability and autism, Fragile X syndrome (FXS) is the most frequent form, for which there is currently no cure. In most FXS patients, the *FMR1* gene is epigenetically inactivated following the expansion over 200 triplets of a CGG repeat (FM: full mutation). *FMR1* encodes the Fragile X Mental Retardation Protein (FMRP), which binds several mRNAs, mainly in the brain. When the FM becomes methylated at 10–12 weeks of gestation, the *FMR1* gene is transcriptionally silent. The molecular mechanisms involved in the epigenetic silencing are not fully elucidated. Among FXS families, there is a rare occurrence of males carrying a FM, which remains active because it is not methylated, thus ensuring enough FMRPs to allow for an intellectual development within normal range. Which mechanisms are responsible for sparing these individuals from being affected by FXS? In order to answer this critical question, which may have possible implications for FXS therapy, several potential epigenetic mechanisms have been described. Here, we focus on current knowledge about the role of DNA methylation and other epigenetic modifications in *FMR1* gene silencing.

## 1. Introduction

Fragile X syndrome (FXS; OMIM #300624) is the most common monogenic cause of inherited intellectual disability (ID). In the general population, the frequency of FXS males is approximately an estimated 1:4000, and that of affected females is 1:7000 [1]. The prevalence of carrier females, who have a high risk of transmitting FXS to their children, is 1 in 250 or higher [2]. A subsequent meta-analysis reduced the frequency of the FXS males to 1.4:10,000 and that of affected females to 0.9:10,000, maintaining unmodified the prevalence of carrier females (1:291) [3].

The causative gene *FMR1* was cloned in 1991, and it is located on the long arm of the X chromosome (in Xq27.3), where the folate sensitive fragile site FRAXA was originally described in affected individuals [4,5]. The gene contains 17 exons spanning 38 kb of genomic DNA, and it is transcribed in a 4.4 kb mRNA with an open reading frame (ORF) of 1.9 kb, plus several isoforms resulting from alternative splicing [6]. In the 5′ UTR, between the promoter region and the start codon, *FMR1* contains a polymorphic CGG repeat typically composed of 20–40 triplets. Based on the number of the CGG repeat, two major pathological alleles may be recognized: premutation (PM) alleles, which present with a number of triplets between 56 and 200, and full mutation (FM) alleles, for which the CGG repeat expands over 200 units. Cytosines of the FM allele are methylated, including those of the CpG island in the promoter region (methylated full mutation: MFM), thus silencing the gene. Premutation alleles, which are not methylated, confer an increased risk for both Fragile X Tremor Ataxia Syndrome (FXTAS; OMIM #300623), a neurodegenerative form of parkinsonism, and Fragile X Premature Ovarian Insufficiency (FXPOI; OMIM #311360), a causative form of premature menopause (before 40 years of age). Premutated females are also at risk of further CGG repeat expansion, generating FM alleles during oogenesis, and of having FXS children. FXS is almost exclusively caused by an MFM of the *FMR1* gene, with other mutations of the gene being very rare. The DNA methylation follows the CGG expansion, leading to the transcriptional silencing of the gene. As a consequence of both dynamic mutation and epigenetic modification, Fragile X Mental Retardation Protein (FMRP) translation is absent or reduced (loss-of-function mutation) despite the persistence of an intact coding sequence of the gene. The absence of FMRP, an RNA binding protein mainly expressed in the brain, causes FXS. These three different clinical conditions associated with CGG instability are also known as FRAXopathies or/and Fragile X-related disorders [7,8]. FXS is caused by a loss-of-function effect of the fully mutated *FMR1* gene, while FXTAS is mainly associated with a toxic gain-of-function effect of the *FMR1* transcript [9]. An anticipation phenomenon similar to that described in other dynamic mutation disorders, e.g., myotonic dystrophy, has not been described in FXS pedigrees. The effect of the mutation is “all or nothing:” when an MFM is transmitted, no matter how large it is, the classical FXS phenotype will be expressed. However, the number of affected individuals tends to increase over generations. This phenomenon, known as the Sherman paradox, is due to the instability of the gene, expanding from PM to MFM from one generation to the next [10].

The promoter region of the *FMR1* gene contains approximately 56 CpG dinucleotides (CpG island), consensus sites for USF1/USF2 (E-Box) and α-Pal/Nrf-1 transcription factors, which positively regulate *FMR1* transcription [11,12]. Next, there are two GC-boxes binding Sp1/Sp3 proteins that show a direct positive regulation of *FMR1* expression [13]. The *FMR1* promoter region includes three initiator-like (InR) sequences, located about 130 nucleotides (nt) upstream the CGG stretch, lacking the canonical TATA box. The canonical transcription start site of the *FMR1* gene is located approximately 50 nt from the CGG repeat. As the size of the CGG repeat expands into the PM range, the transcription start site is shifted further to the upstream sites [14].

Furthermore, a methylation boundary region is located around 650–800 nucleotides upstream the CGG repeat. Normal alleles present an upstream methylated region and downstream unmethylated CpGs up to the *FMR1* promoter, thus allowing for normal transcription. This methylation boundary is lost in MFM alleles, which are completely methylated in this region. The mouse genome conserves a similar methylation boundary, even if human and mouse genomes are only 46.7% identical in the 5′ region upstream the *FMR1* gene. In the methylation boundary region binding sites for various nuclear proteins are located, including CTCF (CCCTC-binding factor), a well-known insulator. The binding of CTCF to its sites probably prevents the spreading of cytosine methylation towards the *FMR1* promoter [15].

DNA methylation is not the only factor regulating the expression of *FMR1*. Four major levels of epigenetic control should be considered to affect *FMR1* gene transcription regulation: (1) cytosine methylation, (2) histone modifications, (3) chromatin remodeling, and (4) action of RNA transcripts.

The first DNA methylation at CpG dinucleotides constitutes the most important epigenetic mechanism used to transcriptionally silence the gene. This reaction is catalyzed by DNA methyltransferases (DNMTs): DNMT1 is responsible for maintaining DNA methylation by acting with its partner UHRF1 and presents a higher affinity for hemimethylated CpG dinucleotides than for unmethylated DNA [16]. Its activity can be modulated by several post-translational modifications [17], such as the demethylation of lysine 1094 (by the lysine demethylase LSD1) and the methylation of lysine 142 (by the lysine methyltransferase SET7) [18]. Instead, the methylation of both lysines leads to DNMT1 degradation, as detected in Lsd1-deficient mouse embryonic stem (ES) cells in which a global DNA hypomethylation is observed [18]. In addition, serine at position 143 can be phosphorylated by the kinase AKT1, preventing the methylation of lysine 142 and protecting from enzyme degradation [19]. This latter post-translational modification provides a remarkable link between DNMT1 stability and the phosphoinositide 3-kinase (PI3K)–AKT–mTOR pathway, which represents a key pathway in cell cycle regulation. DNMT3A and DNMT3B are *de novo* DNA methyltransferases and establish the normal methylation pattern during embryonic development [20,21]. In the human genome, approximately 70% of promoters have CpG islands, of which 75% are less than 850 base-pairs long [22]. Many of these become methylated during development and are highly regulated during different stages of differentiation, like CpG sites of imprinted genes [23] and the X chromosome inactivation center [24]. In humans, numerous processes of mutagenesis are associated with alterations in DNA methylation, which is involved in the genetic stability of the methylated and unmethylated expanded CGG repeat observed in FXS [25]. Methylated DNA is bound by several methyl binding domain (MBD) proteins, such as MeCP2. In *Fmr1* KO mice the level of MeCP2 protein in the cerebral cortex is increased, contributing to the Fxs locomotor hyperactivity phenotype. In *MeCP2* KO mice, the FMRP levels are reduced, suggesting an interplay between the activities of these two proteins [26]. DNA methylation was once considered a stable modification. For a long time, passive demethylation was seen as the only mechanism to demethylate the cytosines, given that newly incorporated cytosines are not remethylated during DNA synthesis. The discovery of ten-eleven translocation (TET) proteins that actively demethylate DNA reversed that concept. TET enzymes can sequentially convert 5-methylcytosine (5mC) to 5-hydroxymethylcytosine (5hmC), 5-formylcytosine (5fC), and 5-carboxylcytosine (5caC). 5fC and 5caC are non-canonical bases; thus, thymine DNA glycosylase (TDG) can remove them, and new (unmethylated) cytosines can be inserted in their place by base excision repair (BER) enzymes. Active DNA demethylation is now a well-accepted mechanism of demethylation, while other proposed mechanisms are less well-defined [27].

Though the critical mechanism of *FMR1* gene silencing is not yet fully understood, its silencing seems to require all four levels of epigenetic regulation. Silenced alleles derived from FXS individuals present with a heterochromatic, non-permissive structure of the *FMR1* locus, while transcriptionally active alleles, such as those of normal and PM carriers, exhibit a euchromatic, permissive configuration. In addition to cytosine methylation, which is the main factor in blocking transcription, FXS cells display histone modifications. Histone deacetylases (HDACs) catalyze histone 3 (H3) and 4 (H4) hypoacetylation; the enzyme histone methyl-transferase (HMT) Set9, in association with coactivator complexes, demethylates lysine 4 on histone 3 (H3K4) [28,29,30]; Polycomb group proteins cause (a) the tri-methylation of lysine 9 on histone 3 (H3K9me3), which recruits chromodomain proteins (such as heterochromatin protein 1 (HP1) spreading heterochromatin status [31]), and (b) the tri-methylation of lysine 27 on histone 3 (H3K27me3) as the ultimate step in the silencing of the gene [32]. Lastly, the tri-methylation of lysine 20 on histone 4 (H4K20me3) is increased near the CGG expansion [33]. Together, these epigenetic changes characterize the silenced, heterochromatic status of the *FMR1* gene, recognized as the cause of FXS.

Contrariwise, the acetylation levels of H3 and H4 in PM alleles are 1.5–2 times higher than those of normal controls [34] and are associated with an increased transcription, with CGG repeats that turn out to exclude nucleosomes both in vivo and in vitro [35]. Though the epigenetic modifications described in PM alleles apparently confer a more open chromatin structure to the *FMR1* locus, *FMR1* mRNA containing premutated CGGs tends to form secondary structures such as hairpins. These structures may cause the stalling of the 40 S ribosomal subunit, contributing to the translational deficit in PM alleles [36] and favoring the use of additional transcriptional start sites [14]. The transcriptional hyperexpression of PM alleles and the lack of expression of MFM alleles remains an unsolved paradox. The existence of unmethylated full mutation (UFM) alleles further complicates this picture. These alleles have been identified in rare individuals with apparently normal intelligence belonging to FXS families [37,38,39]. These individuals are carriers of a CGG sequence expanded in the range of FM that, however, is completely devoid of the cytosine methylation at the CpG island of the *FMR1* promoter region. Histone marks are similar to those of normal alleles: H3 and H4 are acetylated, trithorax group enzymes tri-methylate H3K4 (H3K4me3); H3K27 is di-methylated (H3K27me2), while H3K9 remains partially methylated. These UFM alleles may represent the epigenetic status of FXS cells before *FMR1* gene silencing, at around 11 weeks of gestation [40]. A summary of epigenetic marks in *FMR1* alleles, including UFM carriers, is reported in Table 1.

Despite the great amount of information about the *FMR1* gene generated since its discovery in 1991, further evidence regarding epigenetic mechanisms that lead to its transcriptional silencing is needed. Chromatin remodeling factors are likely to be involved. One of these is CTCF, which binds to the *FMR1* locus using four different binding sites [41,42]. CTCF binds these sites in UFM cells as it does in normal control cells, while this binding is absent in FXS cells [42]. CTCF depletion is accompanied by a reduction of *FMR1* transcription, with no changes in the DNA methylation status of the *FMR1* promoter, both in normal and UFM cell lines [42]. Another interesting candidate protein is the CGG-binding protein 1, CGGBP1 [43], which specifically binds unmethylated CGG repeats in a length-dependent manner [43]. Though its depletion does not change *FMR1* transcription levels [44], its overexpression could be associated with an increased binding to longer CGG repeat tracts that may guide local H3K9 methylation.

## 2. Transcription at the *FMR1* Locus and R-Loops

The fine epigenetic control exerted on the *FMR1* gene is indicative of a locus exposed to a complex transcriptional regulation. Most of our genome is not translated into proteins, while noncoding RNA (ncRNA) transcripts are particularly abundant in the brain [45]. Some of these are about 200 nucleotides long (lncRNAs: long ncRNAs) and are involved in the epigenetic control of the locus from which they arise. The transcription of lncRNAs may occur from both strands. Sense and antisense transcripts are also described for the *FMR1* locus. The most abundant antisense transcript, known as *FMR1-AS1*, is not expressed in FXS cells and is overexpressed in PM cells, exactly like the sense transcript [41]. *FMR1-AS1,* consistently expressed at high levels in the brains of unaffected individuals, undergoes alternative splicing, polyadenylation, and exportation to the cytoplasm. Its transcription arises from two alternative promoters: one overlaps with the *FMR1* bidirectional promoter and the other is mapped in the second intron of the *FMR1* gene. Notably, the latter promoter is mostly used by PM and UFM cells, and it consequently spans the CGG expansion [42]. Furthermore, a region of 9.7 kb corresponding to intron 1 of the *FMR1* gene is spliced in normal and PM alleles using a non-consensus CT to AC splicing site. A specific alternative splicing of intron 2, also demonstrated in PM alleles, arises from a non-consensus CT–AC site. Interestingly, a homologous transcript to human *FMR1-AS1* is present in the mouse genome, suggesting a conserved cellular function for the antisense transcript [41].

An additional ncRNA, known as *FMR4*, was identified by Khalil and colleagues. *FMR4* arises in the antisense orientation with respect to the *FMR1* locus and is particularly expressed in adult frontal cortex and hippocampus. Again, its expression is similar to that of *FMR1* mRNA: FXS alleles do not transcribe *FMR4*, while PM alleles present slight overexpression compared to normal levels of transcription. Transfection experiments were performed in HEK293 cells to functionally correlate both transcripts: siRNAs against the *FMR4* transcript and against *FMR1* mRNA were transfected to downregulate transcription, while vectors that included their sequence were employed to overexpress them. None of these transfection experiments showed modifications of expression of the other transcript and/or functional correlation between both transcripts [46].

*FMR5* and *FMR6* represent two additional ncRNAs arising from the *FMR1* locus [47]. *FMR5* is a lncRNA transcribed 1 kb upstream of the *FMR1* transcription start site in the sense orientation. Its transcriptional start site maps in the methylated region upstream of the methylation boundary. It is expressed in several human brain areas of unaffected and FXS individuals, as well as PM carriers. *FMR6* is transcribed from the antisense strand overlapping with the 3′ UTR and the last three exons of the *FMR1* gene. It is silenced in FXS and PM carriers. Moreover, *FMR6* splices the two introns present between the exons 15–16 and 16–17, as for *FMR1* mRNA. The splicing mechanism recognizes non-canonical consensus sites, as documented for the splicing of *FMR1-AS1* [42].

The antisense transcripts may represent intermediaries in the epigenetic regulation of the *FMR1* transcription, though their exact role has not yet been clarified. For other loci lncRNAs or antisense transcripts induce the assembly of proteins involved in heterochromatin organization or direct the recruitment of silencing machinery [48]. One could hypothesize that Polycomb group proteins (PcGs) are targeted to the CGG repeat of the *FMR1* gene through lncRNAs arising from the locus itself, acting either *in cis*, as demonstrated for the *Kcnq1-Kcnq1ot1* gene cluster [49], or *in trans*, as described for the *Hox*-*HOTAIR* gene clusters [50]. This hypothesis is supported by the evidence that most PcG proteins target G–C-rich regions, so the CGG•CCG-repeats of the *FMR1* gene may be silenced by these complexes.

Recently, it has been demonstrated that specific ncRNAs bind to DNMT1; these RNAs have been defined as DNMT1-interacting RNAs, depicting a new class of transcripts [51]. A genome-wide study that led to the identification of these RNAs started from the molecular analysis of the *CEBPA* gene, involved in hematological malignancies. The DNA methylation levels of the *CEBPA* locus and the transcriptional levels of an lncRNA of *CEBPA*, known as *ecCEBPA (extra coding CEBPA*), are inversely correlated. It was demonstrated that *ecCEBPA* transcript interacts with DNMT1, preventing *CEBPA* methylation and promoting *CEPBA*-mRNA transcription. This functional DNMT1–RNA interaction involves several loci, including *FMR1* [51]. These results suggest that DNMT1 is sequestered by RNAs that act as a shield, preventing the methylation of the locus by DNMT1. To propose a targeted therapeutic approach for FXS it will be relevant to focus our attention on understanding interactions between epigenetic enzymes, such as DNMT1, and transcripts stemming from the *FMR1* locus. Understanding the mechanisms that regulate the interaction between *FMR1* locus and DNMT1 could uncover new targets for potential treatments. A representative model of DNMT1 interaction with DNA and transcription at the *FMR1* locus is reported in Figure 1.

Recently, Cheung and colleagues showed that RNA:DNA hybrid formation promotes transcription by preventing methylation-induced silencing. The authors studied the *BAMBI* gene, which is a negative regulator of transforming growth factor β (TGF-β). They observed a decrease in R-loop formation that exposes the *BAMBI* promoter to the DNA methylation induced by DNMT1. After *BAMBI* silencing, the TGF-β signaling pathway is activated [52]. When the transcript anneals during or after transcription to the template DNA of its genomic locus of origin to form an RNA:DNA hybrid, the non-template single-stranded DNA (ssDNA) is displaced. These atypical structures with three strands of nucleic acids are called R-loops, and they are most abundant in genomic regions enriched in repetitive elements. Genome-wide methylation data showed that the levels of DNA methylation near human promoter regions and transcriptional levels are directly correlated [53,54]. These findings suggest that the start of transcription at CGI promoters is critical for preventing DNA methylation, although the exact mechanism(s) by which this protection takes place remains unclear [55]. Recent data showed that R-loops can have an epigenetic positive role in the *FMR1* locus [56]. R-loops typically originate from guanine-rich (G-rich) regions during gene transcription; they form when an RNA strand invades double-stranded DNA and anneals to the antisense template DNA strand. This causes a Watson–Crick RNA:DNA hybrid and displaces the single stranded DNA (ssDNA) that is not hybridized to RNA [57]. R-loop formation is affected by three main factors—a high density of G, negative supercoiling, and DNA nicks [58]—that influence each other and are not mutually exclusive. DNA:RNA hybridization between nascent RNA and template DNA strand may be favored by both G-clusters and nicks downstream from the promoter on the non-template strand, whereas high G density and negative supercoiling may enhance RNA:DNA hybrid extension and stabilization after its formation [59]. At a reduced G density, R-loop elongation becomes less favorable, leading to the termination of the structure [60]. R-loops are abundant within human gene promoters and terminators, where RNA transcription and processing occur [56,61]. Several studies have demonstrated that an excess or lack of R-loops leads to pathogenic conditions, such as immunodeficiency diseases and neurodegenerative disorders [53,62,63]. Interestingly, it was demonstrated that CGG expansions at the *FMR1* locus are associated with an increase of R-loop formation detected by a S9.6 anti-RNA:DNA hybrid antibody that has the ability to specifically bind R-loops. Through a DOX-inducible episomal *FMR1* vector, it was secondarily demonstrated that R-loop formation is directly proportional to transcriptional activity. Taken together, these experiments support the assumption that there is a direct correlation between CGG repeat expansions and R-loop formation at the *FMR1* locus [64]. Recently, it was also demonstrated that FXS cells accumulate double strand breaks (DSBs), which co-localize with R-loop-forming sequences. Exogenously expressed FMRP in FXS fibroblasts reduces R-loop-induced DSBs, suggesting that FMRP is a genome maintenance protein that prevents R-loop accumulation [65].

## 3. hESCs and iPSCs as Developmental Model for FXS

To further explore how the lack of FMRP causes the FXS phenotype, several FXS animal models have been developed, such as those of mouse [66], rat [67], zebrafish [68], and fruit fly [69]. *Fmr1*-knockout models mimic some biochemical, histological, and behavioral findings of FXS patients. Though in vivo studies offer an important experimental tool for characterizing the cellular roles of FMRP, to date there have been no animal models that recapitulate the human process of CGG expansion and methylation. Human embryonic stem cells (hESCs) are a powerful tool in disease modelling thanks to their ability to recapitulate the early stages of human development [70]. However, the availability of FXS hESCs still represents a limiting factor for researchers. Indeed, the origin of hESC is totally dependent on the rare resource of embryos deriving from preimplantation genetic diagnosis (PGD) [71].

hESCs show a unique epigenetic pattern critical for their self-renewal and pluripotency, as they display a largely permissive chromatin configuration with low levels of compact heterochromatin [72]. Furthermore, pluripotency-associated genes, such as Oct4 and Nanog, are mostly unmethylated, and many promoters of developmental transcription factors are characterized by distinctive histone modification signatures that combine active and repressive epigenetic marks. This bivalence guarantees epigenetic plasticity, allowing embryonic stem cells to rapidly regulate gene expression in response to appropriate environmental signals [73]. Similar to pluripotency genes, during differentiation, the fully mutated *FMR1* gene undergoes a heterochromatinization process, which involves Polycomb complex, G9a protein, and specific DNA methylation that seems to be guided by a Dicer-mediated siRNA mechanism. Following Dicer cleavage, local methyltransferase SUV39H, which mediates the H3K9me3, is responsible for the heterochromatinization of the *FMR1* locus [74]. However, the exact details of the epigenetic mechanism of *FMR1* inactivation during development are not yet understood.

Chorionic villi samples from full mutation fetuses were tested for *FMR1* expression, and the presence of FMRP was characterized at different stages of gestation [40]. Apparently, FMRP is normally expressed until 13 weeks of gestation, and its expression level is similar to that observed in control fetuses. However, FMRP is unexpressed in fetal tissues although the genetic background of CVS and fetal tissues is identical. These data suggest that the hypermethylation of the *FMR1* locus is responsible for the repression of *FMR1* transcription and that methylation occurs early in embryogenesis, at around 11 weeks of gestation [40].

In line with these findings, in hESC lines obtained from a preimplantation FXS embryo [74], *FMR1* showed normal levels of transcription and translation, which means that the promoter region was unmethylated and exhibited active chromatin features.

Upon the spontaneous differentiation of FXS-hESC lines, a downregulation of *FMR1* mRNA level was observed. This decline occurred with the targeted multistep process of heterochromatinization involving firstly histone H3 methylation at lysine 9 or 27 and then DNA methylation [75]. Figure 2 reports the DNA methylation status and the transcriptional activity of the *FMR1* locus in hESC and in corresponding differentiated cells. In a recent study [76], *FMR1* silencing was monitored during in vitro neuronal differentiation over 60 days. Neurons derived from FXS hESCs showed a progressive decrease of the *FMR1* mRNA level until it was totally absent after 51 days post neuronal induction. In particular, in differentiated FXS cells, the *FMR1* promoter showed an increase of the repressive H3K9me3 mark, related to loss of the active H3K4me3 mark. Moreover, it was demonstrated that *FMR1* mRNA was directly involved in silencing the *FMR1* promoter [76]. Briefly, the *FMR1* mRNA includes the transcribed CGG-repeat sequence as part of the 5′ UTR and may hybridize with the complementary CGG tract of the *FMR1* gene to form an RNA:DNA duplex. Using a small molecule, labeled 1a, the interaction between mRNA and the CGG-repeat portion of the *FMR1* gene was found to be disrupted, thus preventing promoter silencing. These results provide a link between trinucleotide-repeat expansion and transcription, as well as a form of RNA-directed gene silencing mediated by RNA:DNA hybrid formation. A direct role of active transcription in the epigenetic silencing of the *FMR1* gene was proved by the knockdown of *FMR1* mRNA in FXS hESCs, where *FMR1* gene is expressed and its promoter region retains histone marks typical of transcriptionally active chromatin. Epigenetic machinery may be triggered at the locus by secondary structures, such as hairpin, formed by CGG repeats in mRNA. The small molecule 1a may unfold and linearize these secondary structures, thus binding the repeating G–G internal loops in the RNA hairpin and inhibiting its thermal melting, preventing *FMR1* silencing [76]. However, hESCs employed in this relevant work appear partially methylated, and the mechanism described by the authors does not explain the existence of UFM alleles, which maintain the *FMR1* transcription in the presence of CGG expansion in the range of FM.

In the same year, Avitzour et al. (2014) established and characterized various full mutation FXS hESCs lines from affected embryos [77]. In some of the FXS hESCs lines, the *FMR1* promoter is partially methylated, ranging from 24% to 65%. These methylation levels remain stable in culture and are directly related to an increase in heterochromatin epigenetic marks. As observed in previous studies [71], in completely unmethylated FXS hESCs, the *FMR1* expression level was identical to that of control hESC lines. The results obtained via the bisulfite DNA sequencing of single colonies suggest a wide variability in methylation levels of full mutation *FMR1* alleles. It follows that clones among the same cell lines were a mixture of cells that showed either the hypermethylation or hypomethylation of the *FMR1* promoter. This is in line with the existence of methylation mosaicism among affected individuals [78]. Furthermore, in FXS hESCs lines, the hypermethylation process, when it occurs, is irreversible, thus leading to a permanent inactivation of *FMR1* transcription, whereas in originally unmethylated FXS hESCs lines, *FMR1* is repressed during differentiation.

Further support to these findings came from the study of other FXS hESC lines containing a combination of larger methylated and smaller unmethylated full mutation alleles. However, after extended propagation in culture, it was observed that the unmethylated alleles were totally methylated and no *FMR1* mRNA was produced [79].

An alternative and valid approach to modelling early stages of FXS pathogenesis is represented by iPSC lines. Though iPSCs closely resemble embryo-derived hESCs, the *FMR1* gene remains inactive and shows histone modifications typical of inactive heterochromatin. This observation further confirms that *FMR1* silencing occurs early during embryogenesis, and it highlights the epigenetic differences between ESCs and iPSCs in FXS modelling [80,81]. While FXS hESCs can recapitulate early embryonic developmental stages, FXS iPSC lines are a useful model to characterize the neuronal defects found in FXS patients and to identify promising new drugs [81].

If the expansion of >200 CGGs represents the main condition triggering *FMR1* gene methylation and silencing, how can the existence of UFM carriers be explained? Considering that FXS patients produce transcripts and R-loops at the first stage of development when the transcription is still active, which are the differences in epigenetic regulation between FXS and UFM individuals? A recent study conducted on iPSCs derived from two unrelated UFM individuals showed that lack of methylation of the *FMR1* promoter is maintained during the neuronal differentiation of these cells. However, *FMR1* became silenced in a small proportion of iPSC clones, with CGG expansion exceeding 400 repeats, thus indicating that the threshold for DNA methylation in UFM cells is higher than the classical 200 CGG repeat. The methylation of large CGG expansion (over 400 repeats) suggested that UFM individuals preserve the intrinsic ability to methylate their mutated *FMR1* gene, but they require larger CGG repeat expansions than those observed in FXS patients. Vice versa, very few iPSC clones derived from FXS patients became unmethylated and transcriptionally active at the *FMR1* locus [82]. In another study, reprogramming iPS cells derived from the fibroblasts of a different UFM individual resulted in a methylated and silenced *FMR1* gene, possibly due to an artefact of the technical procedure [83].

Recent works employed the CRISPR/Cas9 (clustered regularly interspaced short palindromic repeats/CRISPR associated protein 9) system to reactivate *FMR1* transcription, selectively targeting the epigenetic modifications at the *FMR1* locus [84,85,86,87]. Two studies edited the CGG repeats in FXS hESCs and iPSCs. In the first study, an sgRNA (single guide RNA), complementary to the DNA target located at the 5′ UTR of the *FMR1* gene, generated a single double strand break (DBS) [84]. In the second study, sgRNA oligos were designed to target either side of the CGG repeat tract of the *FMR1* gene, resulting in two flanking DSBs [85]. In both cases, the cut site was repaired through non-homologous end joining (NHEJ) activity induced by the Cas9 nuclease. As a result, undifferentiated cells showed a CGG repeat deletion that was followed by the demethylation of the *FMR1* promoter, the upregulation of active histone marks, and the expression of the gene. These changes were also maintained throughout differentiation into mature neurons [85]. Elsewhere, instead of DSB-based editing, a methylation editing approach was used to reverse the excessive methylation of CGG repeats in the *FMR1* gene [86]. In this work, an FXS iPSC model was used to recapitulate the hypermethylation of CGG repeat expansion and epigenetic silencing of *FMR1*. The results showed that the specifically targeted demethylation of CGG repeats by fusing the Cas9 nuclease to the catalytic domain of the TET DNA demethylase was sufficient to reactivate *FMR1* and to partially recover FMRP expression. Upon *in vitro* differentiation, the reactivation of *FMR1* was maintained in edited neurons, which displayed a normal *FMR1* expression and restored electrophysiological properties [86]. Haenfler et al. (2018) used an alternative strategy to reactivate *FMR1* based on transient transfection. The authors combined the Cas9 nuclease with VP16 transcriptional activation domain in order to robustly enhance *FMR1* transcription and increase FMRP levels [87] (Figure 3). 

Taken together, these results suggest that dynamic epigenetic changes in undifferentiated cells demonstrate the possibility of using multiple complementary strategies to obtain the reactivation of the *FMR1* locus. Moreover, the CRISPR/Cas9 editing system could eventually lead to a therapeutic approach in patients with FXS, albeit with caveats regarding the safety and accuracy of this method.

## 4. Epigenetic Approach to Treat FXS

Two different approaches can be proposed to treat FXS: (1) to compensate for the lack of FMRP, one can intervene on pathways in which the protein is involved; (2) to reactivate the expression of the otherwise intact *FMR1* gene, one can act on the epigenetic mechanisms causing its transcriptional inactivation. In CNS cells, and specifically at the synapses, FMRP mainly acts as protein synthesis repressor. The effect of a lack of FMRP on mGLuR signaling [88] has prompted several clinical trials with the purpose of compensating for the lack of FMRP. A promising trial was carried out with AFQ056, a selective inhibitor of mGluR5 with a presumptive epigenetic effect [89,90]. However, a subsequent larger trial was discontinued due to a lack of confirmatory evidence. The existence of the rare UFM individuals, in which CGG expansion remains unmethylated and transcriptionally active, suggested an alternative epigenetic strategy to cure FXS by restoring *FMR1* expression. The epigenetic approach is based on the plasticity of epigenetic signatures, which may be reverted from heterochromatin markers in FXS-silenced alleles to more permissive markers typically described in transcriptionally active cells. Cytosines can be demethylated using 5-azacytidine (5-azaC) or, more efficiently and specifically, 5-aza-2-deoxycytidine (5-azadC). This latter compound is an analog of deoxycytidine and is incorporated into DNA during cell divisions blocking DNA methyltransferases, particularly DNMT1 [91]. Our group firstly obtained the in vitro reactivation of the *FMR1* gene by treating FXS lymphoblastoid cells with 5-azadC [92]. The reactivation was documented by the reappearance of both *FMR1*-mRNA and, in a fraction of treated cells, FMRP. The observed discrepancy between mRNA and protein levels could have been due to the lower translational efficiency in presence of CGG expansion. A synergistic effect on *FMR1* reactivation was obtained by the combined treatment with 5-azadC and several HDAC inhibitors such as butyrate, phenylbutyrate, and trichostatin A. Treatments with HDAC inhibitors alone did not induce any significant reactivation of the *FMR1* gene [93]. Evidence emerging from epigenetic treatments suggests that DNA methylation is prevalent over histone acetylation levels at the *FMR1* locus [94], as was also reported for other heavily methylated genes [95]. Additionally, romidepsin and vorinostat, two FDA-approved HDAC inhibitors, have been tested for their ability to reactivate the *FMR1* gene. Like other HDAC inhibitors, these drugs weakly reactivate the *FMR1* gene in FXS cells; in a few cells, they induce a modest reactivation, while other cell lines exhibit no reactivation. Moreover, most inhibitors of HDAC enzymes are cytotoxic, thus limiting their use in clinical trials for FXS [96].

Treatments of FXS cells with 5-azadC have shown that this compound is able to induce increased histone acetylation and H3K4 methylation levels, while H3K9 methylation is only partially reduced [97]. These epigenetic marks are those typical of a euchromatic configuration of the *FMR1* promoter, effectively converting a methylated FXS into an UFM. Notably, the effect of 5-azadC on genomic DNA methylation is not indiscriminate; rather, it is restricted almost exclusively to the *FMR1* promoter. To support this point, using a microarray screening of 10,814 transcripts, Suzuki et al. (2002) showed that only 51 genes are transcriptionally upregulated after 5-azadC treatment and 23 genes are overexpressed after trichostatin A treatment [98]. We also demonstrated that the reactivating effect of 5-azadC on the mutant *FMR1* gene lasts for several weeks in vitro [99]. A relevant criticism resulting from clinical application of 5-azadC is its toxicity. The use of both 5-azaC and 5-azadC in the treatment of hematological malignancies is generally well-tolerated [100], although their effects over a long time period are not known. Another obstacle to the clinical treatment of FXS with 5-azadC is represented by the evidence that this compound requires cell division, while neurons, the main target cells of the treatment, normally do not divide. Interestingly, there is evidence that 5-azadC effectively induces a reduction of the maintenance DNA methyltransferase DNMT1 through proteasomal degradation, requiring minimal incorporation into DNA [101]. Furthermore, the treatment of FXS-iPS cells and their derived neurons with 5-azacytidine causes a robust *FMR1* reactivation [102]. Though preliminary, this study demonstrated that a targeted epigenetic intervention is feasible. However, the major concern about the clinical use of these epigenetic drugs, like 5-azadC or HDAC inhibitors, is again derived from their mechanism of action that may be extended to the whole genome with unspecific and off-target effects.

Further experiments have corroborated the feasibility of an epigenetic intervention to reactivate the mutant *FMR1* gene. Valproic acid (VPA), an FDA-approved drug to treat seizures and mood disorders, demonstrated a (modest) reactivating effect on *FMR1* gene by acting as histone acetylator without a DNA demethylation effect [103]. Clinical trials based on the use of VPA showed a decrease in hyperactivity disorder in FXS patients, thus further encouraging epigenetic intervention in this genetic condition [104]. A significant reduction of hyperactivity disorder was obtained in a previous clinical trial with L-acetylcarnitine (ALC) [105], a natural compound that may directly increase histone acetylation levels by acting as a donor of acetyl groups. Unfortunately, when ALC is used alone, it is not able to reactivate the *FMR1* gene in vitro, as demonstrated for other similar compounds [93].

In a recent paper, an in vivo platform of FXS iPSCs and differentiated FXS transplants was developed to screen small-molecules and to quickly evaluate the ability of tested compounds to target *FMR1* inactivation [106]. *FMR1*-reactivating compounds are evaluated for their additive and long-term effects. Higher *FMR1* reactivation levels and lower DNA methylation levels of the *FMR1* promoter in FXS transplants were obtained by the combined use of 5-azadC and DZNep (an SAH hydrolase inhibitor) than by treatment with 5-azadC alone. Interestingly, a small reduction of *FMR1* methylation levels was obtained by the use of DZNep alone, which may indirectly inhibit histone modifications. Unlike the chemicals used in other studies [100,107], the reactivating effect of DZNep appears to be maintained for a prolonged period of time. These combined treatments with different epigenetic compounds could reduce the 5-azadC concentration required for *FMR1* reactivation, thus diminishing the potentially toxic side effects of this demethylating agent used in monotherapy. A platform for the analysis of *FMR1*-reactivating treatments in vivo was also developed on two humanized animal models carrying human FXS-affected cells: one implied the generation of a differentiated human FXS transplant in immunocompromised mice and the second involved the transplantation of a specified population of human FXS neural progenitor cells into mouse brains. In both cell types, 5-azadC administration was able to restore *FMR1* expression, representing a proof of principle for *FMR1* demethylation in an in vivo model [106]. Lastly, small molecule inhibitors of H3K9 methylation, such as chaetocin, BIX01294, and DZNep significantly delay the re-silencing of 5-azadC-reactivated *FMR1* alleles. These results also demonstrated that H3K9 methylation precedes DNA methylation and that the removal of DNA methylation with 5-azadC is necessary to obtain the optimal effect of HMT inhibitors on *FMR1* gene expression [108].

## 5. Concluding Remarks

The monogenic nature of FXS makes this rare genetic disorder more amenable than others to pharmacological or molecular therapy. In fact, virtually all FXS patients have the same dynamic mutation, which does not involve the open reading frame of the gene. The epigenetic regulation of *FMR1* suggests the reversible nature of Fragile X syndrome. A substantial body of work has already helped to clarify the pathogenic mechanism that underlies FXS, although further experiments are necessary to elucidate the fine epigenetic regulation of the *FMR1* gene. Again, the phenotype is mainly characterized by neurodevelopmental disorder with ID and behavioral problems ranging from mild to moderate without any structural defects of tissues or organs. However, the effective and stable correction of a genetic defect continues to be a stimulating challenge, even in light of encouraging results obtained by the above-mentioned clinical trials in different cohorts of FXS patients. For future therapeutic approaches, it will be necessary to have a deeper understanding of the pathophysiology underlying this condition.

## Figures and Tables

**Figure 1 biomolecules-11-00296-f001:**
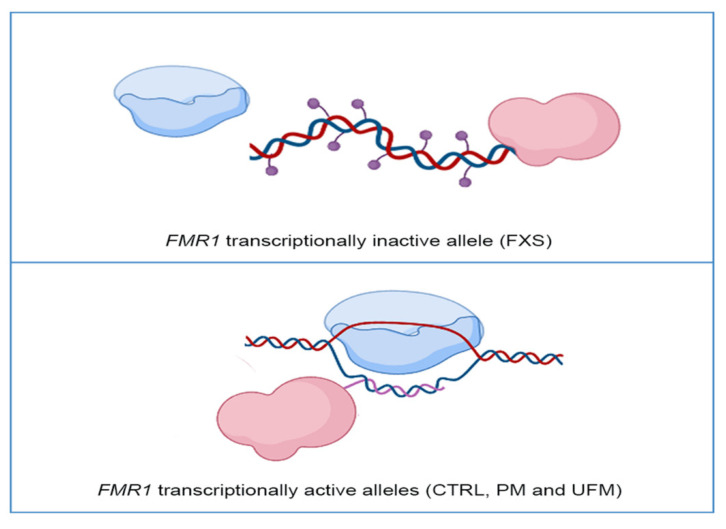
Representative model of DNA methyltransferase 1 (DNMT1) interaction with the *FMR1* locus: In *FMR1* transcriptionally inactive allele, as in Fragile X syndrome (FXS) individuals, DNMT1 (in pink) freely binds DNA and methylates it (violet dots), preventing transcription through RNA polymerase (in blue) (top panel). In *FMR1* transcriptionally active alleles, namely CTRL, PM and UFM, RNA polymerase transcribes RNA (pink line) and DNMT1 binds to nascent RNA and R-loop structures, preventing DNA methylation (bottom panel). Created with BioRender.com.

**Figure 2 biomolecules-11-00296-f002:**
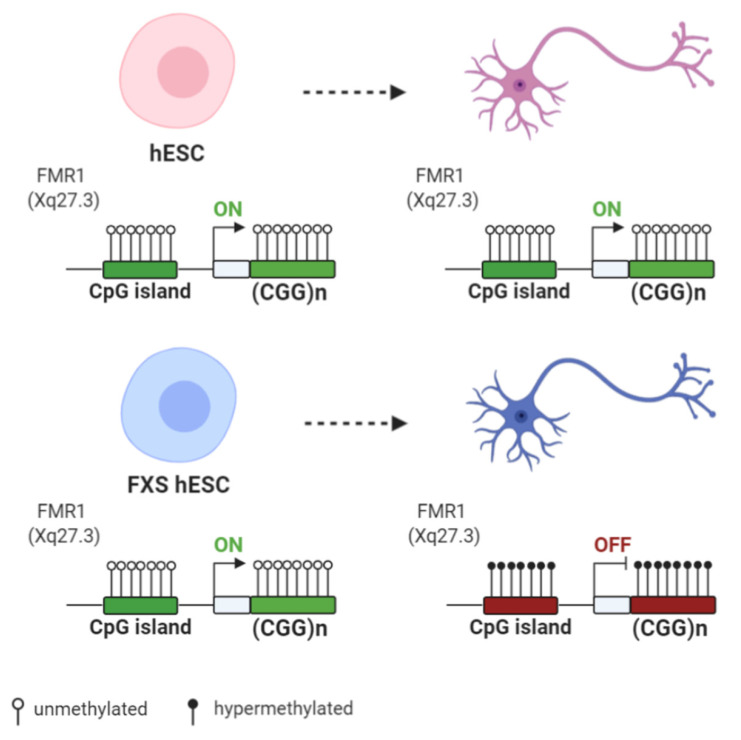
Differences of *FMR1* methylation during differentiation of human embryonic stem cells (hESCs) derived from normal and FXS individuals: Both normal and FXS hESC have an unmethylated *FMR1* promoter and similar *FMR1* expression levels. Upon differentiation, the *FMR1* promoter and the CGG expansion become hypermethylated in FXS hESCs, with the subsequent decrease of the *FMR1* expression level. Created with BioRender.com.

**Figure 3 biomolecules-11-00296-f003:**
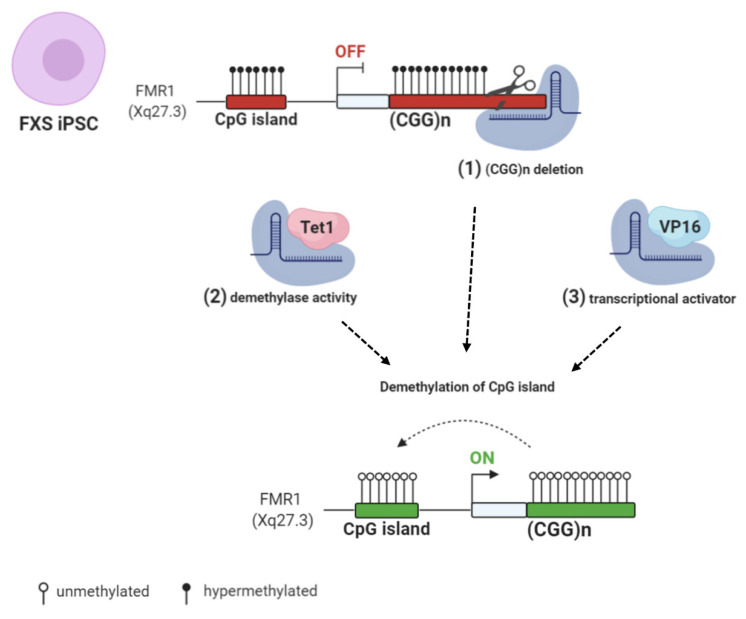
CRISPR/Cas9 (clustered regularly interspaced short palindromic repeats/CRISPR associated protein 9) genome and epigenome editing strategies: (**1**) FXS iPSCs were edited with CRISPR/Cas9 to remove the CGG repeats in the *FMR1* gene [84,85]. (**2**) Targeted demethylation of CGG repeats was achieved by Cas9-Tet1 [86]. (**3**) Transcriptional activator VP16 fused to Cas9 robustly enhanced *FMR1* transcription [87]. All these CRISPR/Cas9 strategies resulted in *FMR1* transcription and Fragile X Mental Retardation Protein (FMRP) production by edited FXS cells. Created with BioRender.com.

**Table 1 biomolecules-11-00296-t001:** Main epigenetics modifications in different *FMR1* alleles. PM: premutation; MFM: methylated full mutation; UFM: unmethylated full mutation; H3: histone 3; H3K9me3: tri-methylation of lysine 9 on histone 3.

	Normal	PM	MFM	UFM
DNA methylation	−	−	+	−
H3–H4 acetylation	+	+ +	− −	+
H3K4me3	+	n.d. *	-	+
H3K9me3	−	n.d.	+ +	+/−
H3K27me3	−	n.d.	+	−
H3K27me2	+	n.d.	−	+
H3K20me3	−	n.d.	+	n.d.
Transcription	+	+ +	−	+ +

* n.d., not determined. “+” means presence and “−” means absence of each single epigenetic modification.

## Data Availability

No new data were created or analyzed in this study. Data sharing is not applicable to this article.

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
