# Peer review of "DNA Methylation, Mechanisms of *FMR1* Inactivation and Therapeutic Perspectives for Fragile X Syndrome"

_biomolecules, 2021, doi:10.3390/biom11020296_

Round 1
Reviewer 1 Report
This is a very nice review on epigenetics of the Fragile X Syndrome and the FMR1 gene. Epigenetics holds a high therapeutic promise in this disease and thus it is an important topic. The review is well organized and can be followed without great efforts. Figures are clear and self explanatory as is the table and provide added value to the manuscript. The authors also detail therapeutic approaches to reactivate the gene through epigenetic modification of the full mutation. This is also a nice review of the state fo the art.
I do not have any major criticism except that the article needs an extensive editing of the English language with special attention to verbal tenses.
Author Response
We wish to thank the Reviewer for his/her kind comments. According to his/her suggestions, we extensively edited the English language.
Reviewer 2 Report
Summary
The present review displays significant and mostly up-to-date insight into the molecular mechanisms underlying FXS with helpful and attractive illustrations and with a good match between title, abstract and content. It suffers from structural and substantial language problems that need to be remedied.
Major Comments
There are many language errors (listed under “Minor Comments” below) and many missing commas (not corrected where not absolutely critical to understanding), addition of which would aid understanding. In total this adds up to a major concern and to a genuine irritant while reading what would otherwise be a convincing and authoritative text. In order to become acceptable, the text will still require editing for language, by a native speaker or an editing service, from Line 230 onwards, where I had to stop detailed corrections because of time constraints.
The article has an overall plagiarism percentage of 32% (see PDF attachment, if provided by the journal), which is excessively high, even though several of the original sources have co-authorship from the current authors. Many verbatim stretches of text, such as those apparently taken from e.g. Doerfler and Böhm 2016 and Vershkov et al. 2019, should be suitably rephrased, and all extensively used source material should be given the courtesy of a formal citation.
It is unfortunate that an article on FXS and underlying mechanisms does not mention the term “anticipation” by name, or the corresponding phenomenon that pedigrees of FXS typically show an increase of severity and penetrance and a decrease in age of onset for the disease. For readers unfamiliar with FXS and its pioneering role in medical genetics, both term and phenomenon should have (at least) a brief honorary mention in the introductory sections.
Line 53 “are also at risk of transmitting their PM allele in the MFM form and thus, of passing on FXS” -> “are also at risk of CGG repeat expansion and creation of FM alleles during oogenesis, and thus of passing on FXS” [The text is imprecise as to the inheritance of the full mutation and moreover suggests transgenerational epigenetic inheritance of the MFM form. Unless you cite a reference for the latter, consider the suggested rewrite.]
Lines 82f You do not make it clear to the reader through formatting or structure whether your subsequent text will systematically deal with the points (1) through (4) raised here. That is what one would expect, but instead points (1) through (3) are expanded without visible separation in section “1. Introduction,” while section (4) is covered in section “2. Transcription…” You should step back and analyse your content and then help the reader by creating a correspondence between structure and content, be it by pointing out that point (4) has its own chapter and/or be it e.g. by using corresponding paragraph headings or corresponding paragraph beginnings, such as “Regarding cytosine methylation, …”; ”Regarding histone modification, …” etc.]
A related minor point: “(1)…(4) RNA transcripts” -> “(1)…(4) action of RNA transcripts” [The other items in the list are processes instead of entities. On an aside, given a clearer structure of the subsequent text and subsequent clarification of the point it is not of great concern that at this stage in the text it is unclear whether with "RNA transcripts" you mean the locus-derived FMR1 mRNA, other RNA from the same locus or trans-acting RNA from other loci.]
Line 157 “Its depletion…” is a confusing section, because you put forward CGGBP1 as an interesting candidate without any qualifying phrases. The reader has to go back to your original publication to understand what you mean to say: “Although its depletion does not change … [44], its overexpression may be associated with an increased binding to longer CGG repeat tracts to regulate local H3-K9 methylation.”
Figure 3 The figure shows the single-DSB CRISPR/Cas9-based strategy of reducing the CGG repeat, but does not explain the underlying mechanism of reduction; neither is this done in the main text. Either display the more intuitive double-DSB strategy in the figure, or briefly state in main text or legend how a single DSB can result in reduced CGG repeat length. Also, strategies (2) and (3) are not genome but epigenome editing strategies; the figure title should therefore be rephrased “CRISPR-Cas9 genome editing” -> “CRISPR-Cas9 [or: CRISPR/Cas9] genome and epigenome editing,” and either legend or main text should point out that mitotically heritable epigenome changes can only be achieved with a combination of epigenome modifications as detailed e.g. in PMID 27662090. Otherwise, strategy (1) is permanent and potentially curative, whereas strategies (2) and (3) are only temporary.
Overall Of 106 total references, only a single reference is from 2020. The review does not claim to cover “latest developments” but it does refer to “present knowledge” in the abstract, which would call for an incorporation of significant recent articles dealing with methylation, mechanisms of FMR1 inactivation and therapeutic perspectives in FXS and any topics specifically covered in the article. Such recent articles include (but are not limited to):
- PMID 33084871, showing in in vivo models, including for FXS, that reduction of MeCP2, an mRNA target of FRMP, in turn reduces FMRP, and that MeCP2 overexpression leads to FMRP increases and amelioration of disease phenotype.
- PMID 33027907, showing differential FRMP expression in FXS and normal urine-derived epithelial cells, as a novel patient-specific cell model for FCS.
- PMID 32966779, showing co-localization of replication-stress-induced DSBs with R-loop sequences and FMRP-mediated amelioration of DSB formation and, by inference, of R-loop accumulation.
- PMID 32269714, showing differential regulation and biological relevance of specific miRNAs in FXS pathology. See also PMID 32107003 for miR-315 action on dFMRP in Drosophila.
- PMID 32230785, evaluating combination therapy of chaetocin, BIX01294 and DZNep with AZA for the prevention of H3K9 methylation and the reactivation of FMR1.
Line 484f The premise of the final sentence does not hold for treatments based on genome editing. I suggest something like the following as a compromise. “Potentially curative approaches based on genome editing of the CGG repeat are costly and not yet ready for clinical application to FXS. Any alternative future therapeutic approaches, however, will critically depend on broader basic knowledge…”
Minor Comments
Owing to the high number of minor comments, I will not proceed to double-check every minor comment and apologise for any inadvertently introduced errors of my own.
Line 18 “(FMRP) that” -> “(FMRP), which”
Line 19 “gestation, FMR1” -> “gestation, the FMR1”
Line 21 “uncommon males have been identified” -> “there is a rare occurrence of males“
Line 24 “by the FXS” -> “by FXS”
Line 34 “estimated in 1:4000” -> “an estimated 1:4000”
Line 35 “females in 1:7000” -> “females 1:7000”
Line 35 “which present high risk” -> “who have a high risk”
Line 35 “successive” -> “subsequent”
Line 37 “1,4:10.000” and “0,9:10.000” -> “1.4:10,000” and “0.9:10,000” (English number format)
Line 39 “The syndrome is a monogenic cause of ID.” (Remove the sentence as redundant and include e.g.“is a monogenic disease and [the most common]” in the first sentence in Line 33.)
Line 39 “1991 and it is” -> “1991 and is” OR “1991, and it is”
Line 61 “while a toxic gain-of-function effect of the FMR1 transcript is mainly associated to FXTAS” -> “while FXTAS is mainly associated with a toxic gain-of-function effect of the FMR1 transcript” [The parallel construction aids understanding, and “associated with” is the correct preposition. Incidentally, instead of reference to gain- and loss-of-function effects, clear reference to absent and excess/aberrant FMR1 mRNA, respectively, might be more helpful.]
Line 42 “DNA and it is transcribed” -> “DNA and is transcribed” OR “DNA, and it is transcribed”
Line 47 “when CGG repeat” -> “for which the CGG repeat”
Line 47 “over200” -> “over 200”
Line 70 “shifts to the upstream sites” -> “is shifted further away from the upstream sites” [Is that what you mean to say?]
Line 72 remove double space
Line 74 “region. Mouse genome” -> “region. The mouse genome”
Line 78 “known insulator. Binding” -> “known insulator sites in the region. Binding”
Line 87 “high affinity […than]” -> “higher affinity […than]”
Line 94 “an remarkable” -> “a remarkable”
Line 99 “became” -> “become”
Line 104 “For long time passive” -> “For a long time, passive”
Line 107 “sheds light to active DNA Demethylation” -> “sheds light on active demethylation” [Also, decide and apply consistently throughout the document whether you want to hyphenate “de-methylate” and derived words or not.]
Line 110 “are replaced” -> “can be inserted in their place” [I needed to step back to understand this sentence.]
Line 134 “subunit contributing to the translational deficit in PM alleles [35] and thus, favoring” -> “subunit, contributing to the translational deficit in PM alleles [35] and thus favoring”
Line 140 “methylation as the” -> “methylation at the”
Table 1 For consistency with the content (e.g. FXTAS and FXPOI are not indicated), the heading “FXS” should be “MFM” or at least “MFM (FXS). Also, use the same spelling as in the main text for H3-K4 and H3-K9 etc.
Line 149 “evidence are needed about” -> “evidence is needed regarding”
Line 150 “if there are” -> “regarding”
Line 152 “bound” -> “binds” [Or change “is” to “was” in the next line]
Line 156 “[43], that” -> “[43], which”
Line 158 “associated to” -> “associated with”
Line 162 “Most part of our” -> “Most of our”
Line 165 “LncRNAs” -> “lncRNAs”
Line 167 “Most relevant” -> “The most relevant”
Line 168 “and it is known FMR1AS1” -> “and is known as FMR1AS1”; “from antisense strand respect to” -> “from the antisense strand with respect to”
Line 171 “undergoes to alternative” -> “undergoes alternative”
Line 175 “Furthermore, 9.7 kb region” -> Furthermore, a region of 9.7 kb” OR “Furthermore, a 9.7-kb region”
Line 176 “using non-consensus” -> “using a non-consensus”
Line 177 “intron 1, non-consensus” -> “intron 1, a non-consensus”
Line 179 “genome stemming from the antisense” -> “genome transcribed in antisense”
Line 182 “orientation respect with the FMR1” -> “orientation with respect to the FMR1”
Line 183f “do not transcribed FMR4” -> “do not transcribe FMR4”; “slightly overexpression” -> “slight overexpression”
Line 190 “from FMR1” -> “from the FMR1”
Line 194 “and last” -> “and the last”
Line 209 “defined DNMT1-interacting” -> “defined as DNMT1-interacting”
Line 210 “of CEBPA gene” -> “of the CEBPA gene”
Line 211 “of CEBPA locus” -> “of the CEBPA locus”
Line 217 “targeted-therapeutic approach” -> “targeted therapeutic approach”
Line 217f “will be relevant focus our attention on understanding interaction” -> “will be relevant to focus our attention on understanding interactions”
Line 220 “a new focus” -> “new targets” [One cannot “uncover” a “focus”; also: avoidance of repeating “focus” in two consecutive sentences]
Line 224 “with FMR1 locus” -> “with the FMR1 locus”
Figure 1 “transcriptionally inactive FMR1 allele” and “transcriptionally active FMR1 allele” would be the correct word order for figure panels and legend. Incidentally, it is more customary to label both panels with (A) and (B), respectively, for reference in the legend and to place the panel title at the top of the panel content.
Line 234 Correct word order of the sentence.
Line 244f “they form when an RNA strand overcomes double-stranded DNA and anneal” -> “they form when an RNA strand [invades/recognizes/disrupts/disanneals] double-stranded DNA and anneals” [If that is what you mean, otherwise rephrase accordingly in order to define “overcomes.”]
Line 252 “When G-rich content decreases R-loops elongation” -> “At reduced G density, R-loop elongation” [If that is what you mean.]
Line 271 “ability of recapitulate” -> “ability to recapitulate”
Line 272f “…dependent on the rare resource of embryo derived by in vitro fertilization that undergone to preimplantation genetic diagnosis (PGD)” [Rephrase to aid comprehension.]
Line 291 “FMRP protein resulted to be unexpressed in fetal tissues” [Rephrase to aid comprehension.]
Line 301f “in hESC and differentiated cells derived by these” -> “in hESC and in corresponding differentiated cells, respectively.”
Line 346 “Despite” -> “Although” [Or change “resemble” to “resembling”.]
Line 356 “derived by” -> “derived from”
Line 358f “few iPSC clones with CGG expansion exceeded 400 repeats, FMR1 became silenced” -> “FMR1 became silenced in a small proportion of iPSC clones with CGG expansions exceeding 400 repeats”
Line 367 Define CRISPR-Cas9 (or CRISPR/Cas9) at first occurrence. I also suggest changing “CRISPR-Cas9” to “CRISPR/Cas9” throughout, as the more common form in the literature and because it facilitates “CRISPR/Cas9-based” and other hyphenated adjectives.
Line 381 “reactivate gene silencing” -> “reactivate gene expression” OR “abolish gene silencing”
Line 396 “reactivate gene silencing, whose coding sequence is intact” -> “reactivate expression of the otherwise intact FMR1 gene” [Otherwise, what does “whose” refer to?]
Line 399 “tempted” -> “prompted”
Line 409 “deoxycitidine” -> “deoxycytidine”
Line 463f “Unlike studies” / “this last platform” -> “Unlike chemicals used in other studies” / “DZNep” [Wrong word connection, where “studies” cannot be compared to “this last platform” and where platform is moreover an inappropriate term for a small-molecule drug.]
Line 466 [The first part of the sentence is only comprehensible with context. I assume that “compared to monotherapy based on 5-azadC alone, combination therapy with other reagents for FMR1 reactivation would reduce the required 5-azadC concentration and any corresponding toxic side effects.”]
Line 478 “Many evidence well clarify the pathogenic mechanism” -> “A substantial body of work has already helped clarify the pathogenic mechanism” [Or rephrase otherwise to make your meaning clear.

Author Response
Major comments.
- We apologize for the grammatical errors and inaccuracies of the English language, that were extensively corrected.
- Some sentences have been rephrased according with the reviewer’s suggestion.
- We agree with the reviewer and now we mentioned in the Introduction the concept of “anticipation” in terms of the so-called “Sherman paradox”.
- We rephrased the concept in line 53 according to the reviewer’s suggestion
- The single points are now better detailed in the text, as suggested by the reviewer.
- We better specified point 4, as correctly suggested.
- Line 157 was modified as suggested.
- Both the legend and the text (from line 371 and on, now 1280 and on) were rephrased to better explain the different mechanisms that underline the three models of genome and epigenome editing summarized in Figure 3. More precisely, the CGG deletion obtained by the two strategies described in refs. n. 85 (single editing) and n. 86 (double editing), respectively, and DNA demethylation of the FMR1 gene obtained by epigenome editing using TET1 (ref. n. 87) were all maintained in hiPSCs/hESCs throughout differentiation into neurons. Contrariwise, reactivation of the FMR1 gene obtained by VP16 domain combined with Cas9 nuclease (ref. n. 88) is transient.
- Some references suggested by the reviewer have been included and mentioned in the text, renumbering them in the exact order.
- We emphasized the potential therapeutic approach based on genome editing at the end of Paragraph 3 and not in the last paragraph.
Minor comments.
We tried to meet all minor comments of the reviewer as he/she suggested point by point.
Reviewer 3 Report
Fragile X syndrome (FXS) is the most common form of monogenic intellectual disability. The molecular cause of the disease is the presence of a dynamic mutation in the FMR1 gene, which leads to the transcriptional silencing of the gene, and thus to a significant reduction or lack of the FMRP protein. While the cause of the disease is known, there are still many questions about how exactly this process works and how it is regulated. The Authors of the manuscript entitled „DNA methylation, mechanisms of FMR1 inactivation 3 and therapeutic perspectives for Fragile X syndrome” clearly and concisely presented the current state of knowledge on the silencing of the FMR1 gene in patients with FXS. What I consider a great asset of the work is presentation of the potential therapeutic possibilities for FXS based on demetylation process.
The title is appropriate for the content of the text.
Authors in some place use word demethylation in other de-methylation. I think the form of this word should be unified.
The description of Table 1 should be more detailed. I miss a description of what the abbreviation n.d. and sign * means.
Line 47 Lack of space „over 200”
Line 54 I suggest using other mutations instead of nucleotide variants
Author Response
- The use of the word demethylation and/or de-methylation has been unified in “demethylation”.
- The description of the abbreviation “n.d.” was included below Table 1.
- Both suggestions (in lines 47 and 54) were changed accordingly.
Round 2
Reviewer 2 Report
Summary
The present review reads well and displays significant and up-to-date insights into the molecular mechanisms underlying FXS with helpful and attractive illustrations and with a good match between title, abstract and content. It should make a frequent point of reference.
I thank the authors for the many diligent corrections made and for the inclusion of updated references. Below, I suggest further minor corrections; however, the manuscript is fully acceptable without my seeing them implemented in a revised version of the manuscript.
Minor Comments
Line 39 “(FMRP) which” -> “(FMRP), which” [Mind the comma; this is a non-defining relative clause. This point occurs several times over in the manuscript, but in order not to be (too) tedious, I only remark this here for the abstract.]
Line 71 “expending” -> “expanding”
Line 80 “range, transcription” -> “range, the transcription”
Line 133 “an euchromatic” -> “a euchromatic” [The following sound and not the spelling determines choice of an/a.]
Line 144 “status [31]; b) tri-methylation” -> “status [31] and b) tri-methylation”
Line 145 Delete “is mediated by”
Line 151 “to excluding nucleosomes” -> “to exclude nucleosomes” [The newly introduced edit is not correct; if a different meaning is intended, consider revising the sentence.]
Line 289 Remove the full stop.
Line 292 “exist” -> “exists”
Line 310 “an unique” -> “a unique” [The newly introduced edit is not correct. The following sound and not the spelling determines choice of an/a.]
Line 336 “with targeted” -> “with the targeted”
Line 358 EITHER “mechanisms… do not” OR “mechanism… does not”
Line 394 [This was fine pre-edit. If you want to use the passive form, write “how can the existence of UFM carriers be explained?” to correct the word order.]
Line 407 “showed methylated” -> “resulted in a methylated”
Line 412 “Basically, in the first” -> “In the first”
Line 414 “More precisely, sgRNA” -> In the second study, sgRNA”
Line 421 “A similar gene editing approach” -> “Elsewhere, instead of DSB-based editing, a methylation editing approach”
Line 475 “suggest” -> “suggests”
Line 491 “effect o 5-azadC” -> “effect of 5-azadC”
Line 492 Delete “before”
Line 495 “effects over a long term period” -> EITHER “effects long-term” OR “effects over a long time period”
Lines 504-506 [I would not encourage the removal of these lines. The assessment here is so positive that the very real limitation of toxicity pointed out in lines 481-482 otherwise inadvertently fades into the background. This also forms a pointer to lines 524 onwards, where combination therapy potentially reduces toxicity compared to monotherapy.]
Line 543 “to a pharmacological treatment” -> “to pharmacological or molecular therapy”
Author Response
In this second revision of the manuscript, we met all minor comments suggested by reviewer #2.
We wish to thank the reviewer #2 for his/her thorough revision, kind comments and for contributing substantially to the improvement of this MS.